# The correlation between pregnancy-related low back pain and physical fitness evaluated by an index system of maternal physical fitness test

**Longfeng Zhou**[1], **Xiaoyi Feng**[1], **Ruimin Zheng**[2]*, **Yuhan Wang**[3], **Mengyun Sun**[2], **Yan Liu**[1]

1 Institute of Physical Education and Training, Capital University of Physical Education and Sports, Beijing, China, 2 National Center for Women and Children's Health, Chinese Center for Disease Control and Prevention, Beijing, China, 3 Department of Pharmacology, Addiction Science, and Toxicology, College of Medicine, University of Tennessee Health Science Center, Memphis, Tennessee, United States of America

* zhengruimin@chinawch.org.cn

**Data Availability Statement:** All relevant data are within the paper.

## Abstract

To investigate incidence of pregnancy-related low back pain (LBP), evaluate physical fitness objectively during pregnancy and analyze the correlation between LBP and physical fitness of pregnant women, 180 pregnant women including 101 in mid-gestation (14–28 gestational weeks) and 79 in late-gestation (28–37 gestational weeks) were recruited and self-reported their LBP. The aerobic ability such as cardiorespiratory fitness and anaerobic ability including strength, endurance, speed, flexibility, and balance were evaluated by a novel materal physical fitness test system. The correlation between LBP and each component in physical fitness test system was analyzed in SPSS. As the results, 135 out of 180 participants (75% of total) had pregnancy-related LBP. Physical fitness of participants in late-gestation was significantly weaker including weaker back strength (p<0.05), less resistance band pull-backs in 30s (p<0.01), less stretching in sit-and-reach test (p<0.001), shorter duration in left legged blind balance test (p<0.05) and weaker bird dog balance(p<0.05) than those in mid-gestation. Correlation analysis indicated that LBP was negatively associated with standing heel raises in 20s (p<0.01) and standing glute kickbacks in 30s (left p<0.01, right p<0.05). Thus, it is concluded that LBP is in high prevalence throughout the entire pregnant course. The pregnant women are prone to have weakened strength of core muscle groups and poorer flexibility and balance along the pregnancy. In addition, their LBP was negatively correlated to strength of back muscle groups of lower limbs.

## 1 Introduction

Pregnancy-related symptoms are experienced among a majority of pregnant women and may reduce the quality of life [1, 2] and adversely affect mental health, birth outcome and postpartum recovery [3–5]. Low back pain is one of the most common symptoms in pregnancy [6]. Statistics suggest that more than 50% of women experience low back pain during pregnancy

**Funding:** This study was supported by grant from the National Key Research & Developmental Program of China (2022YFC2703800).

**Competing interests:** The authors have declared that no competing interests exist.

[7], and more than 30% of pregnant women will continue to be affected in the postpartum period [8]. Considering the safety of fetal development, pregnant women prefer conservative treatments such as support belt use, kinesio taping, activity reduction and even bed rest to relieve low back pain [9–11]. However, use of different support belts [12] and bed rest [13] are proved to have no significant effect on low back pain and the efficacy and safety of kinesio taping for pregnant women is still in debate [11]. Actually, exercise and physical therapy are the first-line treatments as non-pharmacological methods regarding to general low back pain [14]. Increasing evidence shows that low back pain such as sacroiliac joint mechanical pain and herniated discradicular pain are accompanied by spine tenderness and muscular weakness [14]. To treat these types of low back pain, improving physical fitness such as movement control exercise designed to strengthen muscles and improve spinal posture has a positive effect on disability as well as pain alleviation [15].

With growing understanding of safety of exercise during pregnancy, physical activity during pregnancy has been a common sense throughout the world [16–18]. All the pregnant women without contraindications are strongly recommended to have certain exercise to avoid pregnancy-related symptoms according to the WHO guidelines (2020) and other reports [19–21]. In recent years, more and more exercise prescriptions have been designed to treat pregnancy-related low back pain [22–24]. However, a review covering 34 trials examining 5121 pregnant women provided low-quality evidence that any land-based exercise significantly reduced pain [12]. This low-quality evidence and diversity of exercise prescription used bring us back to reconsider the relationship between pregnancy-related low back pain and individual physical fitness.

It is reported that greater self-reported overall physical fitness assessed by International Fitness Scale (IFIS) was associated with pregnancy-related lumbar pain [25]. Meanwhile daily activities based on free descriptive answers including standing up from chair, lying down, tossing and turning were found to be correlated with low back pain during pregnancy [26]. However, it is not well established to evaluate comprehensive physical fitness during pregnancy with objective measurements [27]. Given that the changes of physical fitness level and the morbidity of low back pain along the entire pregnant course are still obscure, it is very difficuilt for most current exercise programs to accurately and specifically address low back pain at different stages of gestational stages.

Our team initially constructed a physical fitness test system for pregnant women which includes physique, physiological function and physical quality and have 23 component tests. These 23 component tests were constructed through three rounds of expert discussion with references to the physiological characteristics of pregnant women, exercise guidelines, exercise contraindications, exercise needs and existing physical fitness tests of people with limited exercise and elderly people. 60 participants completed all the tests without complaints or discomfort reported [28]. This evaluation system is scientific, practical and safe, and can be used as an evaluation tool for physical fitness during pregnancy.

Using this evaluation system, this study aimed to evaluate the physical fitness of pregnant women objectively, investigate pregnancy-related low back pain based on Oswestry Disability Index (ODI) [29] and explore the correlation between the low back pain and those 23 component tests in the physical fitness evaluation system. On one hand, this study provided methodological support for objective measurement of physical fitness for pregnant women. On the other hand, this preliminary exploration of relationship between physical fitness and low back pain during pregnancy laid a foundation for the research on the correlation and causality between physical fitness and other common symptoms during pregnancy, and also provided an objective reference for scientific exercise strategy for pregnant women.

## 2 Materials and methods

### 2.1 Study design

The present study is a part of a project about correlation between objective physical fitness during pregnancy and several common pregnancy-related symptoms including low back pain. The physical fitness evaluation we used here is a test system for pregnant women mentioned above [28]. More details is displayed in Fig 1. To be scientific and reliable, all the participants performed the tests under the guidance of a computerized program with a video showing the standardized movement for every test, an accurate test timer and the same time interval between tests. A professional assistant and a doctor presenting were responsible for safety issues and collecting the test results.

Meanwhile all participants rated their low back pain from 0 to 5 compared with non-pregnant state and the correlation between low back pain and physical fitness was analyzed.

This study was approved by research ethics committee of Capital University of Physical Education and Sports (201910012).

### 2.2 Participants

A total of 180 participants were recruited and classified into mid-gestation group (14–28 weeks) and late-gestation group (28–37 weeks) according to their gestational week on the test day. 84 pregnant women (44 in mid-gestation and 40 in late gestation) were enrolled in July 2021 through Shanxi maternity and child health hospital (Shanxi, China) and 96 pregnant women (57 in mid-gestation and 39 in late gestation) were enrolled in September 2021 through Shandong provincial hospital east center (Shandong, China).

They were evaluated to meet the following criteria: (1) age between 22–40 years old; (2) gestation between 14–37 weeks with single fetus; (3) both pregnant women and fetus in health condition according to prenatal examination; (4) no assisted reproductive technology applied; (5) no smoking or alcohol habits; (6) no history of premature delivery, recurrent abortion, or fetus growth restriction; (7) no cardiopathy, hypertension and cerebrovascular disease history; (8) no pregnancy complications including premature rupture of membranes, polyhydramnios or oligohydramnios, fetal malformations and so on; (9) no dyskinesia or other exercise contraindications. In addition, to confirm that the participants have the ability to complete all tests safely in the assessment, we excluded the participants who had self-scaled low back pain intensity more than 3 based on the Oswestry Disability Index (ODI) [29]. The enrollment of participants whose pain intensity was 3 was determined after medical evaluation and discussion with the doctor and participants.

All the participants signed a written informed consent after being informed our study aims and procedures. All the data and personal information of participants were identified by numbers, and the personal information of the subjects will not be disclosed.

### 2.3 Self-reported low back pain

The participants were asked to scale the low back pain they suffered during pregnancy in comparison with their non-pregnancy state. They rated the intensity of their low back pain as "0 = no pain", "1 = very mild pain", "2 = moderate pain", "3 = fairly severe pain", "4 = very severe pain", "5 = the worst imaginable pain" as the Oswestry Disability Index (ODI) [29].

### 2.4 Physical fitness assessment

All the participants were informed that the physical fitness assessment lasted around 30 minutes. For every component test, there was a teaching video showing the standard movement

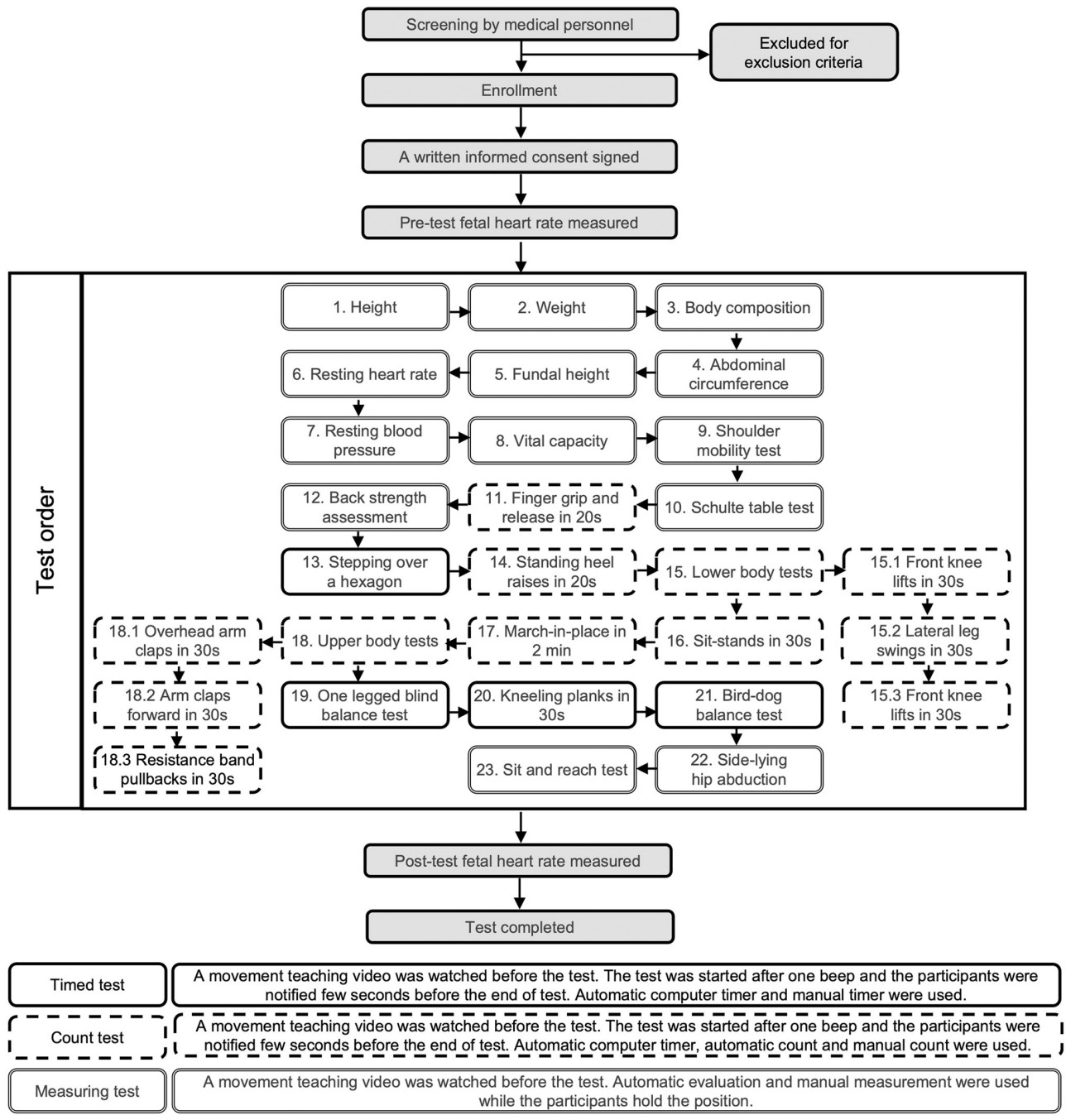

**Fig 1. Flow chart of study participants.**

and the participants were asked to follow the instructions repeating the movement. The repetition in certain duration and/or time spent to complete movement were recorded. Accompanying with medical staff, the participants could terminate the assessment immediately if they felt uncomfortable. They were instructed to wear a wristband with a heart rate sensor. The assessment was started once the heart rate sensor was connected to the heart rate monitor and

their basic information including name, age, height, gestation week was imported into the database in the computer.

A series of tests were performed in the order shown in Table 1. Instruction and equipment used for every test were attached below. The participants were allowed to take a break after all the test completion and asked for the assessment feedback. They also had post-test fetal heart rate collected when their heart rate and breath back to normal.

## 2.5 Statistical analyses

Statistical analyses were performed using IBM SPSS Statistics for Windows (version 26.0, IBM Corp., Armonk, NY, USA) and the statistical significant was set as $P<0.05$. Data from individuals with incomplete results were excluded for analysis. All the data were presented as mean ± standard deviation (SD). Kolmogorov–Smirnov test was used to evaluate the normality of data of every test. Differences of variables between mid-gestation and late-gestation group were conducted by independent sample T test for normal distributed data and Mann-Whitney tests for non-normal distributed data. The relationships between low back pain with all physical fitness component tests were determined by spearman correlation analysis.

## 3 Results

### 3.1 Clinical anthropometric characteristics of participants

180 participants were completed the assessment as our final samples including 101 participants in mid-gestational stage and 79 participants in late-gestational stage. The anthropometric and clinical characteristics of participants were shown in Table 2. With the fetal development, the weight ($P<0.01$), body mass index (BMI) ($P<0.01$), body fat ($P<0.01$), abdominal circumference ($P<0.01$), fundal height ($P<0.01$) and resting heart rate ($P<0.01$) of participants in late-gestation were significantly higher than those in mid-gestation. However, there was no significant difference in age, height, blood pressure and fetal heart rate between mid-gestation and late-gestation groups. Additionally, no difference in fetal heart rate before and after assessment indicates that our physical fitness assessment was safe for the participants and their babies.

### 3.2 Self-reported low back pain during pregnancy

The level of low back pain during pregnancy was self-rated by the participants and shown in Table 3. Generally, 25% of all participants did not have low back pain at all (pain intensity = 0). 57.78% and 16.67% of participants experienced very mild pain that did not affect daily life (pain intensity = 1) and moderate pain that can be tolerated during daily life (pain intensity = 2), respectively (greater score means more severe pain). Only 1 participant suffered from fairly severe pain that may affect daily life (pain intensity = 3). In comparison of pain intensity of participants during middle and late gestations, the proportion of participants who have very mild pain was decrease, while the proportion of those who have no pain and moderate pain were increased indicating that some participants were adopted to the increasing abdominal load but it was even worse in some cases. In addition, the mean pain intensity during mid-gestion and late gestation was 0.91±0.62 and 0.95±0.71, respectively, indicating that low back pain happens throughout the pregnancy course.

### 3.3 Objective assessed physical fitness during pregnancy

All parameters in physical fitness assessment were displayed in Table 4. Compared with women in mid-gestational stage, those in late gestation had significant weaker back strength ($p<0.05$) and less resistance band pullbacks in 30s ($p<0.05$), indicating that the core muscle

**Table 1. Workflow of the physical fitness assessment.**

| Test order | Test name | Movement description | Test goal | Duration (second) | Equipment |
|---|---|---|---|---|---|
| 1 | Height | stand straight with bare feet | Longitudinal growth | 30 | A height meter |
| 2 | Weight | stand on the flatform with bare feet and push the handle till the result displayed | Horizontal growth | 30 | Inbody composition analyzer (InBody370) |
| 3 | Body composition | | % of body fat | 180 | |
| 4 | Abdominal circumference | lay on the exam table after urination | Fetal growth | 30 | A measuring tape and an exam table |
| 5 | Fundal height | | | 30 | |
| | Fetal heart rate monitor before test | | | | |
| 6 | Resting heart rate | have upper arm wrapped by a digital sphygmomanometer at heart level quietly till result displayed | Cardiovascular function | 60 | A digital sphygmomanometer |
| 7 | Resting blood pressure | | | | |
| 8 | Vital capacity | inhale as deeply as possible and then exhale fully in one-time | Pulmonary ventilatory function | 15 | Hand-held spirometer (LK-T2016) |
| 9 | Shoulder mobility test | bend one elbow above and the other below the shoulder to get both fists close to each other behind the body to measure the distance between fists | Mobility of shoulders | 15 | A 50cm ruler |
| 10 | Schulte table test | find, read out and click all numbers in an ascending order of 16 numbers randomly arranged in a 4×4 table to record the time spent | Attention and neuromuscular Coordination | 10 | An app of Schulte table test |
| 11 | Finger grip and release in 20s | make tight fists and releases in 20s to record repetitions | Speed of finger movement | 20 | A stopwatch |
| 12 | Back strength assessment | pull up the handlebar without elbows and knees bent or backward fall standing on the chassis of a back dynamometer | Strength of back muscle group | 30 | back dynamometer (JY-BLJ) |
| 13 | Stepping over a hexagon | step over every line of a hexagon without body turning around and step back to the center in a clockwise order to record time spent | Multidirectional exercise performance | 30 | A roll of floor tape, a measuring tape, a stopwatch |
| 14 | Standing heel raises in 20s | raise heels off and lower down to the floor for 20s to record repetitions | Movement speed of lower limbs | 20 | A stopwatch |
| 15 | Lower body tests | | | | |
| 15.1 | Front knee lifts in 30s | lift a knee to the hip level and release on 30s to record repetitions | Endurance of lower limbs' muscle group | 60 | A stopwatch |
| 15.2 | Lateral leg swings in 30s | swing a leg out forming 45˚ with body and swing back in 30s for each leg to record repetitions | | | A stopwatch and a chair |
| 15.3 | Standing glute kickbacks in 30s | lift a leg behind to touch glute and lower back in 30s for each leg to record repetitions | | | |
| 16 | Sit-stands in 30s | stand up quickly and stably from a chair and sit back for 30s to record repetitions | strength of muscle group in lower limbs | 30 | |
| 17 | March-in-place in 2 min | march in place for 2min with lifted ankle higher than mid-calf of the supporting leg to record step repetitions | Cardiovascular function | 120 | A stopwatch |
| | 1min rest | | | | |
| 18 | Upper body tests | | | | |
| 18.1 | Overhead arm claps in 30s | clap hands over head and lower back to the shoulder height for 30s to record repetitions | Endurance of upper limbs' muscle group | 30 | A stopwatch |
| 18.2 | Arm claps forward in 30s | clap hands forward with elbow straight and abduct arms at shoulder height for 30s to record repetitions | | 30 | |
| 18.3 | Resistance band pullbacks in 30s | hold two ends of a 10-pound resistance band that is fixed in the middle at shoulder height, pull them back to elbow bent at 90˚ and release at shoulder height for 30s to record repetitions | Strength and endurance of upper limbs' muscle group | 30 | A stopwatch, a resistance band (10lb) and a resistance band holder |
| 19 | One legged blind balance test | hold the standing position with one foot off the ground and closed eyes as long as possible to record the time duration till maximum 30s | Proprioception and static balance without vision | 60 | A stopwatch |

(*Continued*)

**Table 1.** (Continued)

| Test order | Test name | Movement description | Test goal | Duration (second) | Equipment |
|---|---|---|---|---|---|
| 20 | Kneeling plank | hold the position lying down with elbows and knees on the mat with raised calves and straight back for as long as possible to record the time duration till maximum 30s | Strength of the core muscle group | 30 | A yoga mat and a stopwatch |
| 21 | Bird-dog balance test | hold the bird-dog position with raised an arm overhead and contralateral leg lifted for as long as possible to record the time duration till maximum 30s | Ability to maintain body balance | 60 | |
| 22 | Side-lying hip abduction | lie down on their side on a yoga mat and have the upper leg abducted as far as they can to measure the angle between both legs | flexibility of hips | 60 | universal protractor (187–153, 0-500mm) |
| 23 | Sit and reach test | Lean forward slowly at the hips to push the cursor by their fingertips with straight legs sitting on a testing mat | | 30 | Sit and Reach Flexibility Assessment Tester (wi99894 China) |
| | Test feedback | | | | |
| | Fetal heart rate monitor after test | | | | |

**Table 2. Clinical anthropometric characteristics of participants (N = 180).**

| | Mid-gestation (N = 101) | Late-gestation (N = 79) | Total (N = 180) |
|---|---|---|---|
| Age | 30.41±3.35 | 29.47±3.46 | 29.99±3.42 |
| Gestational week | 20.94±3.43 | 31.88±2.20** | 25.74±6.19 |
| Height (cm) | 162.65±4.86 | 162.99±5.91 | 162.80±5.33 |
| Weight (kg) | 63.13±8.86 | 68.99±8.56** | 65.70±9.19 |
| BMI (kg/cm$^2$) | 24.00±3.28 | 25.99±2.71** | 24.87±3.19 |
| Body fat (%) | 31.22±5.15 | 33.24±4.38** | 32.11±4.92 |
| Abdominal circumference (cm) | 88.38±8.15 | 97.96±6.40** | 92.58±8.82 |
| Fundal height (cm) | 20.28±3.40 | 30.06±3.02** | 24.58±5.84 |
| Heart rate (bpm) | 89.57±11.93 | 95.20±12.02** | 92.04±12.26 |
| Blood systolic pressure (mmHg) | 107.31±12.09 | 109.41±16.35 | 108.23±14.11 |
| Blood diastolic pressure (mmHg) | 67.57±9.80 | 66.58±8.42 | 67.14±9.20 |
| Fetal heart rate (pre-test) (bpm) | 145.27±6.68 | 144.97±6.21 | 145.14±6.46 |
| Fetal heart rate (post-test) (bpm) | 145.46±6.42 | 145.38±7.44 | 145.42±6.87 |

Data are presented as mean ± SD,

*$p<0.05$,

**$p<0.01$ mid-gestation vs late gestation

Abbreviation: BMI = body mass index

**Table 3. Self-reported low back pain during pregnancy.**

| Low back pain | | | Mid-gestation (N = 101) | Late-gestation (N = 79) | Total (N = 180) |
|---|---|---|---|---|---|
| Pain intensity | 0 (no pain) | N(%) | 23(22.77%) | 22(27.85%) | 45(25%) |
| | 1 (very mild pain) | N(%) | 65(64.36%) | 39(49.37%) | 104(57.78%) |
| | 2 (moderate pain) | N(%) | 12(11.88%) | 18(22.78%) | 30(16.67%) |
| | 3 (fairly severe pain) | N(%) | 1(0.99%) | 0(0) | 1(0.55%) |
| Average pain intensity | | | 0.91±0.62 | 0.95±0.71 | 0.93±0.66 |

**Table 4. Comparison between objectively measured physical fitness during mid-gestation and late-gestation.**

|  | Mid-gestation (N = 101) | Late-gestation (N = 79) | Total (N = 180) |
|---|---|---|---|
| Vital capacity (ml) | 2631.97±609.88 | 2594.75±595.84 | 2615.63±602.36 |
| 2 min march-in-place (steps) | 191.12±20.64 | 190.05±21.64 | 190.65±21.03 |
| Kneeling plank (s) | 29.35±2.52 | 26.32±8.70 | 28.02±6.23 |
| Back strength assessment (kg) | 32.09±10.22 | 28.59±9.07* | 30.56±9.86 |
| 30s sit-stands | 12.49±2.51 | 11.78±2.84 | 12.18±2.67 |
| Schulte table test (s) | 4.33±0.87 | 4.50±0.92 | 4.40±0.89 |
| 20s finger grip and release | 29.74±7.83 | 30.82±8.52 | 30.22±8.13 |
| 20s standing heel raises | 15.53±4.09 | 15.38±4.35 | 15.47±4.20 |
| 30s overhead arm claps | 23.50±4.16 | 23.01±3.96 | 23.28±4.07 |
| 30s arm claps forward | 24.11±3.65 | 23.25±3.70 | 23.73±3.69 |
| 30s resistance band pullbacks | 22.87±4.87 | 21.22±4.59* | 22.14±4.81 |
| 30s left front knee lifts | 24.51±5.62 | 23.76±4.69 | 24.18±5.23 |
| 30s right front knee lifts | 23.59±5.48 | 22.99±4.33 | 23.33±5.01 |
| 30s left lateral leg swings | 25.77±6.80 | 23.95±5.93 | 24.97±6.48 |
| 30s right lateral leg swings | 24.83±6.20 | 23.76±5.19 | 24.36±5.79 |
| 30s left standing glute kickbacks | 24.63±6.51 | 24.49±6.06 | 24.57±6.30 |
| 30s right standing glute kickbacks | 23.89±5.49 | 23.27±5.51 | 23.62±5.49 |
| Left shoulder mobility test (cm) | 12.51±6.33 | 12.73±6.63 | 12.60±6.45 |
| Right shoulder mobility test (cm) | 9.47±5.29 | 9.99±5.91 | 9.70±5.56 |
| Sit and reach test (cm) | -1.80±7.63 | -7.46±9.63** | -4.28±8.99 |
| Left side-lying hip abduction (degree) | 79.93±15.96 | 78.35±16.74 | 79.23±16.28 |
| Right side-lying hip abduction (degree) | 82.78±17.09 | 81.77±15.54 | 82.34±16.39 |
| Stepping over a hexagon (s) | 25.86±4.50 | 26.39±3.72 | 26.09±4.17 |
| Left-legged blind balance test (s) | 7.30±8.04 | 4.78±4.20* | 6.20±6.73 |
| Right-legged blind balance test (s) | 5.56±4.87 | 4.85±5.03 | 5.25±4.94 |
| Left bird-dog balance (s) | 25.17±8.46 | 24.45±8.93 | 24.85±8.65 |
| Right bird-dog balance (s) | 26.25±7.22 | 23.43±9.15* | 25.02±8.22 |

*p<0.05,

**p<0.01 mid-gestation vs late gestation

strength of pregnant women in late-gestation was lower than that in mid-gestation. In addition, participants in late-gestation had less stretching in sit-and-reach test (p<0.01), shorter duration in left-legged blind balance test (p<0.05) and weaker bird-dog balance with right arm and left leg (p<0.05) suggesting that the flexibility and balance of pregnant women in late-gestation was poorer than that in mid-gestation. For the remaining tests, there was no significant difference between middle and late gestations.

### 3.4 Correlations between low back pain and physical fitness during pregnancy

The coefficient of spearman correlation between self-reported low back pain with every component of physical fitness assessment were shown in Table 5. Among all these component tests, the low back pain was negatively associated with standing heel raises in 20s (p<0.01) and standing glute kickbacks in 30s (left, p<0.01; right, p<0.05). During mid-gestation, the low back pain was negatively correlated with standing glute kickbacks in 30s (both left and right, p<0.05).

**Table 5. Correlations between low back pain and physical fitness during pregnancy.**

| | Mid-gestation (N = 101) | Late-gestation (N = 79) | Total (N = 180) |
|---|---|---|---|
| Height (cm) | -0.032 | -0.026 | -0.031 |
| Weight (kg) | -0.083 | -0.083 | -0.042 |
| BMI (kg/m$^2$) | -0.069 | 0.012 | -0.016 |
| Body fat (%) | 0.017 | 0.084 | 0.037 |
| Abdominal circumference (cm) | 0.083 | -0.004 | 0.055 |
| Fundal height (cm) | -0.056 | -0.117 | -0.040 |
| Heart rate (bpm) | 0.090 | 0.087 | 0.081 |
| Blood systolic pressure (mmHg) | -0.044 | -0.127 | 0.036 |
| Blood diastolic pressure (mmHg) | -0.007 | 0.157 | 0.060 |
| Vital capacity (ml) | 0.125 | -0.062 | 0.038 |
| 2 min march-in-place (steps) | 0.065 | -0.103 | -0.014 |
| Kneeling plank (s) | 0.022 | -0.016 | -0.010 |
| Back strength assessment (kg) | -0.100 | 0.101 | -0.013 |
| 30s sit-stands | -0.024 | -0.085 | -0.065 |
| Schulte table test (s) | 0.096 | -0.218 | -0.049 |
| 20s finger grip and release | -0.135 | 0.026 | -0.052 |
| 20s standing heel raises | -0.172 | -0.214 | -0.196** |
| 30s overhead arm claps | -0.098 | 0.034 | -0.036 |
| 30s arm claps forward | -0.144 | 0.050 | -0.049 |
| 30s resistance band pullbacks | -0.138 | 0.137 | -0.010 |
| 30s left front knee lifts | -0.149 | -0.107 | -0.126 |
| 30s right front knee lifts | -0.081 | -0.086 | -0.083 |
| 30s left lateral leg swings | -0.049 | -0.158 | -0.101 |
| 30s right lateral leg swings | -0.062 | -0.104 | -0.077 |
| 30s left standing glute kickbacks | -0.216* | -0.162 | -0.196** |
| 30s right standing glute kickbacks | -0.203* | -0.167 | -0.184* |
| Left shoulder mobility test (cm) | -0.040 | -0.094 | -0.061 |
| Right shoulder mobility test (cm) | -0.026 | -0.050 | -0.329 |
| Sit and reach test (cm) | -0.032 | 0.041 | -0.040 |
| Left side-lying hip abduction (degree) | 0.128 | -0.016 | 0.053 |
| Right side-lying hip abduction (degree) | 0.167 | -0.076 | 0.046 |
| Stepping over a hexagon (s) | -0.117 | 0.111 | -0.013 |
| Left-legged blind balance test (s) | 0.087 | 0.090 | 0.089 |
| Right-legged blind balance test (s) | 0.124 | 0.023 | 0.070 |
| Left bird-dog balance (s) | -0.019 | -0.002 | -0.012 |
| Right bird-dog balance (s) | 0.037 | 0.107 | 0.062 |

*p<0.05,

**p<0.01 significantly correlated with low back pain

## 4 Discussion

In this study, physical fitness and low back pain were evaluated in 180 pregnant women and the correlations between low back pain and tests of physical fitness evaluation system including cardiorespiratory fitness, muscular strength, endurance, speed and body flexibility, balance were explored. As the main results, the physical fitness of pregnancy women was declined as seen weaker core strength, poorer flexibility, and worse balance in late gestation. Low back pain is commonly experienced during pregnancy. Less low back pain was correlated with

greater lower limb motor function. Specifically, low back pain was negatively correlated with standing heel raises in 20s and left and right standing glute kickbacks in 30s.

Low back pain is one of common symptoms during pregnancy [26] as indicated that up to 75% of participants have low back pain in our study. It usually begins in the second trimester and continues in the remaining pregnancy course [6]. Consistently, our study shows that 77.23% of mid-gestational women and 72.15% of late-gestational women experience low back pain. As pregnancy progresses, the pelvis is anterior tilted gradually to compensate the center of gravity shift due to the enlarging abdomen and hormonal changes which accumulates extra load on lumbar spines inducing low back pain [6]. Additionally, lumbar muscles remain contracted to compensate the weak strength of abdominal muscles because of enlarging gravid uterus which contributes low back pain [30]. The average intensity of the low back pain in the participants during mid-gestation and late gestation was 0.91±0.62 and 0.95±0.71, respectively. The pain intensity in late gestation was slightly higher than that in mid-gestation without significant difference. It is consistent with published results [25, 26]. A probability is that pregnant women having more severe low back pain were excluded in our study because of safety consideration.

Pregnant women with low back pain usually have troubles with basic activities of daily living such as walking, standing up from chair and crunching [26, 31]. Our study designed a series of specific motions that are fundamental for daily activities to evaluate physical fitness during pregnancy and compared them between mid-gestation and late-gestation regarding cardiorespiratory fitness, muscular strength, endurance, speed and body flexibility, balance. In all motion tests, similar vital capacity and similar steps in 2 minutes March-in-place test show no difference in cardiorespiratory function between participants in mid-gestation and late gestation. In addition, they have comparable performance in Schulte table test, 20s finger grip and release test, 20s standing heel raise test and stepping over a hexagon indicating that speed ability and agility are not affected as the pregnancy processes. However, participants in late gestation have significant weaker back strength and significant less resistance band pullbacks in 30 seconds. These results suggest that the core muscle strength and endurance of pregnant women in late-gestation was weaker than those in mid-gestation. It seems that, increased relaxin release, a common reason inducing low back pain [32], affects the core skeletal muscular system. In addition, participants in late gestation have poorer flexibility and balance as indicated by less stretching distance in sit-and-reach test, shorter duration in left legged blind balance test and weaker bird dog balance with raised right arm and left leg. Cortell-Tormo JM et al. suggested that improvements in muscular system and balance probably help females experience less low back pain [33]. Combined this finding and the differences between mid-gestation and late gestation in our results, it is clear that pregnancy-related physical fitness decrease may contribute to low back pain during the pregnant course.

In fact, it is already reported that greater overall physical fitness is associated with less bodily pain and lumbar pain during pregnancy [25]. Our study explored the correlation between low back pain and different types of physical fitness. Our result shows that, regardless the gestational week, low back pain in all participants is negatively correlated with standing heel raises in 20s and standing glute kickbacks (both left and right) in 30s. As we know, gluteal muscles and hamstring muscles are the primary muscles engaged in standing glute kickbacks [34], and gastrocnemius muscles and soleus muscle are responsible for standing heel raises [35]. The patients who have low back pain generally have weak gluteal medius [36] and chronic low back pain limits trunk-pelvis, pelvis-thigh coordination in sagittal plane as well as lower extremities during walking [37]. While heel raise and glute kickback in standing posture are commonly repeated in walking, our speculation is that pregnant women with low back pain have weaker lower back and limb kinematics, especially lumber-pelvis-thigh-calf coordination.

The contracted and weakened lumber muscles not only contribute to low back pain as mentioned earlier, but also result in weakened musculoskeletal activities of hip, knees, and ankles which lead to poorer balance and flexibility subsequently.

Interestingly, when we considered the correlations in middle and late gestations separately, low back pain was correlated with standing glute kickbacks (both left and right) in 30s only during middle gestation. Standing glute kickback tests the rapid contraction capability and coordination of muscle groups in hips and back thighs that is relatively complicated and difficult for all the participants. Participants without low back pain in mid-gestation could do better than those with low back pain making it correlated in glute kickback test, however, participants in late gestation have more bulky body and complete kickback with more difficulty no matter whether low back pain is present or not. All the evidence suggest that pregnancy-related low back pain is correlated to weak physical fitness. Further studies are needed to explore preventive and therapeutic exercise treatments for low back pain during pregnancy, especially treatments improving ability of core and lower limbs.

Our study has limitations. Firstly, we have 180 participants for current study. The sample size could be larger. Secondly, all participants enrolled are pregnant. Only difference between mid-gestation and late gestation is analyzed. Participants in preconception and puerperium could also be recruited to evaluate change of physical fitness and low back pain because of pregnancy and delivery since low back pain continues in some women in the postpartum period.

## 5 Conclusion

Low back pain is in high prevalence throughout the entire pregnant course. The pregnant women are prone to have weaker strength of core muscle groups and poorer flexibility and balance along the pregnancy. In addition, their low back pain was negatively correlated to strength of back muscle groups of lower limbs.

## Supporting information

**S1 Checklist. STROBE statement—Checklist of items that should be included in reports of observational studies.**
(DOCX)

## Author Contributions

**Conceptualization:** Longfeng Zhou, Ruimin Zheng.

**Data curation:** Xiaoyi Feng, Yan Liu.

**Formal analysis:** Xiaoyi Feng, Mengyun Sun.

**Funding acquisition:** Ruimin Zheng.

**Project administration:** Longfeng Zhou.

**Supervision:** Longfeng Zhou, Mengyun Sun.

**Writing – original draft:** Longfeng Zhou, Xiaoyi Feng, Yuhan Wang.

**Writing – review & editing:** Longfeng Zhou, Ruimin Zheng, Yuhan Wang.

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
