## [Editor Report · Decision Letter 0]

18 May 2023

PONE-D-23-11001The correlation between pregnancy-related low back pain and physical fitness evaluated by an index system of maternal physical fitness testPLOS ONE

Dear Dr. Zheng,

Thank you for submitting your manuscript to PLOS ONE. After careful consideration, we feel that it has merit but does not fully meet PLOS ONE’s publication criteria as it currently stands. Therefore, we invite you to submit a revised version of the manuscript that addresses the points raised during the review process.

We look forward to receiving your revised manuscript.

Kind regards,

Oluwaleke Ganiyu Sokunbi, PhD

Academic Editor

PLOS ONE

Journal Requirements:

Additional Editor Comments (if provided):

INTRODUCTION

1. A summary of the findings from prevıous studıes on the normal values of some of the physical fitness parameters (age and gender matched) assessed in this present study, wrıtten in a small paragraph withın the ıntroductıon, will be useful

2. Few lines on the author’s perceived sıgnıfıcance of the study could also be ıncluded within the ıntroduction sectıon

METHODS

PARTICPANTS

What was the sampling technique?

Lıne 145-159; PHYSICAL FITNESS ASSSESSMENT

1. What the order of carryıng out the tests that were lısted ın Table 1? –was ıt serially as lısted in the table or randomly?

2. Were the participants allowed any resting perıods ın between one test and the next one? What measure was put in place to prevent carry over effect of performing one test before the next one

3. How accurate are the scores obtained from this tests? How many repetitions of the test were carried out before recordıng the fınal output? Could ıt be once or three tımes with an average scores being calculated?

4. Authors should provıde a reference for the procedure used for each of the test as carried out, perhaps in a separate column in Table 1
---

## [Author Response · Author response to Decision Letter 0]

2 Jun 2023

Dear Dr. Oluwaleke Ganiyu Sokunbi,

Thank you very much for your comments to improve our manuscript. We have revised our manuscript according to the journal requirements and answered your questions. Look forward to more communication with you and reviewers.

Best,

Ruimin Zheng

---

## [Decision Letter · Decision Letter 1]

1 Sep 2023

PONE-D-23-11001R1The correlation between pregnancy-related low back pain and physical fitness evaluated by an index system of maternal physical fitness test

Dear Dr. Zheng,

Thank you for submitting your manuscript to PLOS ONE. After careful consideration, we feel that it has merit but does not fully meet PLOS ONE’s publication criteria as it currently stands. Therefore, we invite you to submit a revised version of the manuscript that addresses the points raised during the review process.

We look forward to receiving your revised manuscript.

Kind regards,

Renato S. Melo, PhD

Academic Editor

PLOS ONE

Reviewers' comments:

Reviewer's Responses to Questions

**Comments to the Author**

1. If the authors have adequately addressed your comments raised in a previous round of review and you feel that this manuscript is now acceptable for publication, you may indicate that here to bypass the “Comments to the Author” section, enter your conflict of interest statement in the “Confidential to Editor” section, and submit your "Accept" recommendation.

Reviewer #1: (No Response)

Reviewer #2: (No Response)

2. Is the manuscript technically sound, and do the data support the conclusions?

Reviewer #1: Partly

Reviewer #2: Partly

3. Has the statistical analysis been performed appropriately and rigorously? 

Reviewer #1: No

Reviewer #2: Yes

4. Have the authors made all data underlying the findings in their manuscript fully available?

Reviewer #1: Yes

Reviewer #2: Yes

5. Is the manuscript presented in an intelligible fashion and written in standard English?

Reviewer #1: Yes

Reviewer #2: Yes

6. Review Comments to the Author

Reviewer #1: This study analyses data from 180 pregnant women at various stages in pregnancy who complete a suite of fitness tests designed by the authors. The details surrounding design of the tests and application in pregnancy is detailed in a separate publication. 5. The prevalence of LBP reported by the women in the study (13% mid-gestation and 23% late-gestation).

There are some grammatical and typos throughout that would need to be addressed but overall the article is well written.

My main concerns are with the statistical analysis and reporting of results:

1. Descriptive statistics for numeric data are reported as mean +- SD without any consideration of outliers or skewness in the data across the variables measured. Were the distributions of the numeric variables examined for outliers or skewness? This should be detailed in the statistical analysis and reported. Were numeric data is strongly skewed it should be summarised using median and IQR and between groups tests would use the Mann-Whitney test.

2. Multiple testing: Table 4 contains 27 variables which are examined using independent samples t-tests. With this many tests, the researchers run a considerable risk of Type 1 errors - that is rejecting a null hypothesis incorrectly. It would be expected to find at least one statistically significant p-value even if there were no differences in the population. Some consideration and adjustment for Type 1 errors due to multiple testing should be made, for example using a Bonferroni adjustment. This also holds for Table 5 as well with 36 different variables tested for association.

3. Table 4 results - could the differences be explained by the extra weight of the baby later in pregnancy? For example the sit and reach test would be inhibited by the size of the bump. The difference in the kneel plank could also be due to the heavy baby. The balance tests appear to be inconclusive (differences for left leg for blind balance and not for right leg, differences for right-bird dog but not left) - again this could be due to running so many statistical tests - these difference might not be strong enough to be interest after adjusting for multiple testing.

3. Correlation Analysis: LBP is measured on an ordinal scale from 0 = no pain, 1 = very mild pain, 2 = moderate pain therefore a Pearson correlation or (Pearson partial correlation if looking to adjust for gestational week) is not appropriate.

An analysis suitable for ordinal data should be undertaken. Even if the Pearson analysis was appropriate, the results do not support the conclusions given in the abstract of strong negative correlations. The strongest correlation is -0.216 for 'right standing kick backs' which would considered a "weak" statistically significant correlation, a moderate/strong correlation would imply a Pearson correlation of at least 0.5 in magnitude.

4. Conclusions drawn from the results: The correlation results are incorrectly reported as "strongly negatively correlated" - this needs to be much more considered after an appropriate analysis of the ordinal pain scale used.

I would strongly recommend the authors work with a biostatistician to ensure the statistical analyses are correctly undertaken with careful considerations of the distribution of the data, outliers, the ordinal nature of the the pain scale and multiple testing as well as ensuring that any conclusions are not overstated.

Reviewer #2: Thank you for the opportunity to review the manuscript “The correlation between pregnancy-related low back pain and physical fitness evaluated by an index system of maternal physical fitness test”. It focuses on an important topic related to a majority of pregnant women. In my opinion the manuscript represents a good research practise. Nevertheless, following major consideration should be addressed:

1. Table 1: please, describe in more detail how the physical fitness tests were chosen into the testing battery. As described in previously, there is a wide variety of fitness testing protocols for pregnant women (see Romero-Gallardo L, Roldan Reoyo O, Castro-Piñero J, May LE, Ocón-Hernández O, Mottola MF, Aparicio VA, Soriano-Maldonado A. Assessment of physical fitness during pregnancy: validity and reliability of fitness tests, and relationship with maternal and neonatal health - a systematic review. BMJ Open Sport Exerc Med. 2022 Sep 23;8(3):e001318. doi: 10.1136/bmjsem-2022-001318. PMID: 36172399; PMCID: PMC9511659.).

2. Can be the results of sit-and-reach test affected by the size of the pregnant belly or it reflects the change in flexibility of pregnant women?

3. What is the validity and reliability of the physical fitness test battery used in the manuscript?

4. Table 3: as low back pain intensity 4 and 5 were one of the exclusion criteria, do not display them at the table. Information about the number of participants prepared to participate in the study and those who were excluded because the high pain intensity would bring an additional valuable information.

English language: As non-native English speaker, I found some language mistakes. Consider language editing of the final manuscript.

7. PLOS authors have the option to publish the peer review history of their article (what does this mean?). If published, this will include your full peer review and any attached files.

Reviewer #1: No

Reviewer #2: No

---

## [Author Response · Author response to Decision Letter 1]

20 Oct 2023

Dear Dr. Renato S. Melo and reviewers,

Thank you very much for your valuable comments and suggestions. We have attached revised manuscript and response to the comments. 

Look forward to your feedback!

Sincerely,

Ruimin Zheng

---

## [Decision Letter · Decision Letter 2]

9 Nov 2023

The correlation between pregnancy-related low back pain and physical fitness evaluated by an index system of maternal physical fitness test

PONE-D-23-11001R2

Dear Dr. Zheng,

We’re pleased to inform you that your manuscript has been judged scientifically suitable for publication and will be formally accepted for publication once it meets all outstanding technical requirements.

Kind regards,

Renato S. Melo, PhD

Academic Editor

PLOS ONE

Additional Editor Comments (optional):

Reviewers' comments:

Reviewer's Responses to Questions

**Comments to the Author**

1. If the authors have adequately addressed your comments raised in a previous round of review and you feel that this manuscript is now acceptable for publication, you may indicate that here to bypass the “Comments to the Author” section, enter your conflict of interest statement in the “Confidential to Editor” section, and submit your "Accept" recommendation.

Reviewer #1: All comments have been addressed

Reviewer #2: All comments have been addressed

2. Is the manuscript technically sound, and do the data support the conclusions?

Reviewer #1: Yes

Reviewer #2: Yes

3. Has the statistical analysis been performed appropriately and rigorously? 

Reviewer #1: Yes

Reviewer #2: Yes

4. Have the authors made all data underlying the findings in their manuscript fully available?

Reviewer #1: Yes

Reviewer #2: Yes

5. Is the manuscript presented in an intelligible fashion and written in standard English?

Reviewer #1: Yes

Reviewer #2: Yes

6. Review Comments to the Author

Reviewer #1: The authors have adequately addressed all previous review comments and recommended changes. The statistical analysis and interpretation of results are now appropriate.

Reviewer #2: The authors have addressed the reviewers' comments properly. I recommend the manuscript for publication.

7. PLOS authors have the option to publish the peer review history of their article (what does this mean?). If published, this will include your full peer review and any attached files.

Reviewer #1: No

Reviewer #2: No

---

## [Editor Report · Acceptance letter]

12 Dec 2023

PONE-D-23-11001R2 

The correlation between pregnancy-related low back pain and physical fitness evaluated by an index system of maternal physical fitness test 

Dear Dr. Zheng:

I'm pleased to inform you that your manuscript has been deemed suitable for publication in PLOS ONE. Congratulations! Your manuscript is now with our production department. 

Kind regards, 

on behalf of

Dr. Renato S. Melo 

Academic Editor

PLOS ONE